# Early Aspirin Discontinuation Following Acute Coronary Syndrome or Percutaneous Coronary Intervention: A Systematic Review and Meta-Analysis of Randomized Controlled Trials

**DOI:** 10.3390/jcm9030680

**Published:** 2020-03-03

**Authors:** Paul Guedeney, Jules Mesnier, Sabato Sorrentino, Farouk Abcha, Michel Zeitouni, Benoit Lattuca, Johanne Silvain, Salvatore De Rosa, Ciro Indolfi, Jean-Philippe Collet, Mathieu Kerneis, Gilles Montalescot

**Affiliations:** 1Sorbonne University, ACTION Study Group, INSERM UMRS_1166, Cardiology Institute, Pitié Salpêtrière Hospital (AP-HP) Paris, 75013 Paris, France; paul.guedeney@aphp.fr (P.G.); mesnier.jules@gmail.com (J.M.); abchafarouk@gmail.com (F.A.); michel.zeitouni@gmail.com (M.Z.); b-lattuca@chu-montpellier.mssante.fr (B.L.); johanne.silvain@aphp.fr (J.S.); jean-philippe.collet@aphp.fr (J.-P.C.); mathieu.kerneis@aphp.fr (M.K.); 2Division of Cardiology, Department of Medical and Surgical Science, Magna Graecia University, 88100 Catanzaro, Italy; sabatosorrentino@hotmail.com (S.S.); saderosa@unicz.it (S.D.R.); indolfi@unicz.it (C.I.)

**Keywords:** aspirin, P2Y_12_ inhibitors, antiplatelet therapy, acute coronary syndrome, percutaneous coronary intervention

## Abstract

The respective ischemic and bleeding risks of early aspirin discontinuation following an acute coronary syndrome (ACS) or percutaneous coronary intervention (PCI) remain uncertain. We performed a prospero-registered review of randomized controlled trials (RCTs) comparing a P2Y_12_ inhibitor-based single antiplatelet strategy following early aspirin discontinuation to a strategy of sustained dual antiplatelet therapy (DAPT) in ACS or PCI patients requiring, or not, anticoagulation for another indication (CRD42019139576). We estimated risk ratios (RR) and 95% confidence intervals (CI) using random effect models. We included nine RCTs comprising 40,621 patients. Compared to prolonged DAPT, major bleeding (2.2% vs. 2.8%; RR 0.68; 95% CI: 0.54 to 0.87; *p* = 0.002; I^2^: 63%), non-major bleeding (5.0 % vs. 6.1 %; RR: 0.66; 95% CI: 0.47 to 0.94; *p* = 0.02; I^2^: 87%) and all bleeding (7.4% vs. 9.9%; RR: 0.65; 95% CI: 0.53 to 0.79; *p* < 0.0001; I^2^: 88%) were significantly reduced with early aspirin discontinuation without significant difference for all-cause death (*p* = 0.60), major adverse cardiac and cerebrovascular events (MACE) (*p* = 0.60), myocardial infarction (MI) (*p* = 0.77), definite stent thrombosis (ST) (*p* = 0.63), and any stroke (*p* = 0.59). In patients on DAPT after an ACS or a PCI, early aspirin discontinuation prevents bleeding events with no significant adverse effect on the ischemic risk or mortality.

## 1. Introduction

The optimal antithrombotic regimen following ACS or PCI has known considerable evolutions over the last thirty years. However, current guidelines still recommend the continuation of prolonged DAPT including aspirin and a P2Y_12_ inhibitor based on ancient pivotal randomized trials [1,2,3]. Since then, implementation of newer generation drug-eluting stents (DES), the widespread use of lipid lowering therapy, and a new generation of P2Y_12_ inhibitors have led to a reduction of ST or non-stent related MI following PCI or ACS [4,5]. In these circumstances, the benefit of sustained DAPT may translate into a smaller absolute ischemic event risk reduction, which might be potentially outweighed by the associated higher risk of bleeding [6]. Since aspirin yields limited additional platelet inhibition when associated with P2Y_12_ inhibitors, aspirin-free strategies have been evaluated in several recent randomized controlled trials enrolling ACS or PCI patients; in some of these studies patients also had an indication for chronic oral anticoagulation (OAC) [7,8,9,10,11,12,13,14,15,16,17,18,19]. Most of these trials (but not all) reported lower rates of bleeding without aspirin, but all of them were underpowered as to properly evaluate the associated ischemic risk. This systematic review and meta-analysis aims to evaluate the safety and efficacy of early aspirin discontinuation with P2Y_12_ inhibitors single antiplatelet therapy continuation, as compared with a strategy of sustained DAPT following an ACS or PCI, in patients with or without concomitant OAC treatment.

## 2. Materials and Methods

### 2.1. Research Strategy and Selection Criteria

In accordance with the PRISMA (Preferred Reporting Items for Systematic Reviews and Meta-analysis) guidelines (Appendix A), we searched PubMed/Medline, CENTRAL (Cochrane Central Register of Controlled trials), clinicaltrials.gov, and slide presentations from the latest international conferences for relevant abstracts and manuscripts published up to 27 September, 2019. The following keywords were used: randomized controlled trial; acute coronary syndrome; percutaneous coronary intervention; antithrombotic therapy; aspirin; clopidogrel; ticagrelor; prasugrel; rivaroxaban; apixaban; edoxaban; dabigatran. Citations were screened at the title and abstract level and retrieved if considered relevant. The inclusion criterion was a RCT with a clinical primary endpoint, comparing a strategy of early aspirin discontinuation (i.e., with aspirin placebo or no aspirin treatment) and P2Y_12_ inhibitors (clopidogrel, ticagrelor, or prasugrel) continuation, to a strategy of prolonged DAPT including aspirin and P2Y_12_ inhibitors, following ACS or PCI, in patients with or without indication for chronic OAC. No restrictions on follow-up or study size were applied. The exclusion criteria were observational study design (including single-arm pilot studies), non-English-language studies, editorial, letters, expert opinions, case reports or series and studies with duplicated data. Two authors independently evaluated studies for eligibility and discrepancies were resolved by a third reviewer. The primary safety endpoint of interest was major bleeding, as defined in each trial (Appendix A). Other safety endpoints of interest were non-major bleeding as well as all bleeding, defined as the composite of major and non-major bleedings. The primary efficacy endpoint was all-cause death. Other efficacy endpoints of interest were MACCE, MI, definite ST, definite or probable ST, any stroke, and ischemic stroke as reported in each trial (Appendix A). The study is registered in PROSPERO (CRD42019139576).

### 2.2. Data Extraction

Relevant data elements, including baseline population and procedural characteristics, were independently collected from each trial into a pre-specified structural dataset. Efficacy and safety endpoints were collected at the longest available time of follow-up according to the intention-to-treat principle. Discrepancies in the data collection were resolved by consensus. The risk of bias of the included studies was assessed according to the Cochrane Collaboration guidelines.

### 2.3. Statistical Analysis

Risk ratios and 95% CI were estimated using Mantel-Haenszel random-effects models according to DerSimonian and Laird. Fixed effect models for all efficacy and safety endpoints were also reported in the Online documents. Heterogeneity among trials for each outcome was estimated with Chi-square tests and quantified with I^2^-statistics. Visual inspections of funnel plot were used to evaluate potential publication bias and small study effect. In order to evaluate the public health impact of the early aspirin discontinuation strategy on safety endpoints, we pooled outcomes data of all studies into a single population to calculate the absolute risk difference (ARD) and the number needed to treat (NNT) to avoid one bleeding event [20]. Sensitivity analyses were pre-specified: (i) evaluating the impact of safety events after exclusion of trials in which background OAC treatment was not homogenous between early aspirin discontinuation and prolonged DAPT groups; (ii) evaluating the safety of early aspirin discontinuation across various bleeding scales; (ii) evaluating the effect of early aspirin discontinuation according to the P2Y_12_ inhibitors predominantly used (i.e., clopidogrel or ticagrelor), and according to the delay before aspirin discontinuation (i.e., one month or three months) in trials without chronic indication for OAC; (iv) evaluating the effect of early aspirin discontinuation using adjudicated data from the GLOBAL LEADERS Adjudication Sub-Study (GLASSY) [21]. A *p*-value <0.05 was considered as statistically significant. Analyses were conducted using Cochrane’s Review Manager (RevMan) version 5.3 (The Cochrane Collaboration, Copenhagen, Denmark).

## 3. Results

### 3.1. Systematic Review

A total of nine RCTs were included in the present meta-analysis (Appendix A), comprising 40,621 patients of whom 20,320 (50%) were treated with a strategy of early aspirin discontinuation. OAC treatment was present in five RCTs, as an inclusion criterion, representing 11,532 (28.4%) patients. Main characteristics of the included trials are detailed in Table 1, baseline patients’ characteristics are detailed in Table 2, and procedural characteristics are detailed in Appendix A. Male and diabetic patients represented 75.6% and 31.7% of the overall population respectively and the index event was an ACS in 52.4% of the cases. Median follow-up was 1 year (range 0.5 to 2 years).

### 3.2. Safety Endpoints

The association of early aspirin discontinuation with safety endpoints is detailed in Figure 1.

The early aspirin discontinuation strategy was associated with a significant reduction of major bleeding (2.2% vs. 2.8%; RR 0.68; 95% CI: 0.54 to 0.87; *p* = 0.002; I^2^: 63%), with an ARD of −0.62% and NNT: 162; non-major bleeding (5.0% vs. 6.1%; RR: 0.66; 95% CI: 0.47 to 0.94; *p* = 0.02; I^2^: 87%), with an ARD of –1.12% and NNT: 89; as well as all bleeding (7.4% vs. 9.9%; RR: 0.65; 95% CI: 0.53 to 0.79; *p* < 0.0001; I^2^: 88%), with an ARD of − 2.57% and NNT: 39. The effect of early aspirin discontinuation was consistent in patients with and without chronic background OAC, without significant interaction for major, non-major and all bleeding outcomes (*p* = 0.78; *p* = 0.31 and *p* = 0.79, respectively).

### 3.3. Efficacy Endpoints

The early impact of early aspirin discontinuation vs. prolonged DAPT on efficacy endpoint is detailed in Figure 2 and Figure 3.

No significant difference between the two strategies was observed with respect to all-cause death (2.6% vs. 2.7%; RR: 0.96; 95% CI: 0.84 to 1.11; *p* = 0.60; I^2^: 13%), MACCE (5.4% vs. 5.3%; RR: 0.97; 95% CI 0.87 to 1.08; *p* = 0.60; I^2^: 28%), MI (2.0% vs. 2.0%; RR: 1.02; 95% CI: 0.88 to 1.19; *p* = 0.77; I^2^: 8%), definite stent thrombosis (0.79% vs. 0.71%; RR: 1.07; 95% CI: 0.81 to 1.43; *p* = 0.63; I^2^: 0%), definite or probable stent thrombosis (0.48% vs. 0.37%; RR: 1.34; 95% CI: 0.68 to 2.62; *p* = 0.40; I^2^: 37%) (Appendix A), any stroke (0.99% vs. 1.03%; RR: 0.94; 95% CI: 0.76 to 1.17; *p* = 0.59; I^2^: 0%), as well as ischemic stroke (0.69% vs. 0.73% RR: 0.97; 95% CI: 0.61–1.53; *p* = 0.89; I^2^: 44%) (Appendix A). The effect of early aspirin discontinuation was consistent in patients with and without chronic background OAC, without any significant interaction for all-cause death, MACCE, definite stent thrombosis, definite or probable stent thrombosis, any stroke, and ischemic stroke (*p* = 0.38; *p* = 0.27; *p* = 0.55; *p* = 0.28; *p* = 0.87 and *p* = 0.85, respectively). There was a significant interaction between patients with and without chronic background OAC for MI (*p* = 0.04).

### 3.4. Sensitivity Analyses and Bias Assessment

Results for the safety and efficacy endpoints remained consistent with the application of a fixed effect model (Appendix A, respectively). The results of early aspirin discontinuation on safety events remained consistent after exclusion of RCT without homogenous background OAC between the experimental and control groups (Appendix A). The association of early aspirin discontinuation with major, non-major, and major or non-major bleeding remained consistent across the various bleeding scales used in each trial (Appendix A). The results of early aspirin discontinuation on safety and efficacy events remained consistent according to the type of P2Y_12_ inhibitors predominantly used (i.e., clopidogrel vs. ticagrelor) (Appendix A) or the duration of DAPT prior to aspirin discontinuation (i.e., 1 month vs. 3 month DAPT duration) (Appendix A), in trials where patients had no indication for chronic OAC. Finally, the interaction between patients with or without OAC with respect to the association of early aspirin discontinuation with MI remained significant when using adjudicated MI from the GLOBAL LEADERS Adjudicated Sub-Study (GLASSY) trial (Appendix A), but was no longer significant when using site-reported MI (Appendix A). No evidence of publication bias or small study effect was found for both safety and efficacy outcomes (Appendix A). Internal bias assessment for each study is reported in Appendix A.

## 4. Discussion

The choice, at the individual level, of the optimal antiplatelet strategy following ACS or PCI is a conundrum that requires stratifying both ischemic and bleeding risks. Our analysis, based on very recent randomized trials, comprising a total of 40,621 patients, demonstrates that early discontinuation of aspirin following ACS or PCI in patients with or without concomitant OAC treatment is associated with a significant reduction of major, non-major and all bleedings (Graphical abstract). This improved safety is not associated with any significant difference of all-cause death, MACCE, MI, definite ST, definite or probable ST, any stroke, or ischemic stroke.

Historically, aspirin is the first line antithrombotic treatment in cardiovascular diseases [8]. Subsequently, novel antiplatelet agents, including P2Y_12_ inhibitors, were evaluated on top of aspirin, in pivotal RCTs [9,22]. Various DAPT regimens, based on aspirin with more or less prolonged duration of more or less potent P2Y_12_ inhibitors, have been evaluated to further reduce the ischemic residual risk, following ACS or PCI [23,24,25]. However, any reduction of the thrombotic risk has usually been offset by an increased risk of bleeding [22,24]. Of importance, bleeding following PCI has been associated with all-cause mortality and is thus paramount to prevent [26].

In particular, patients with AF requiring oral anticoagulants, presenting with an ACS or undergoing PCI, are exposed to a considerable increased risk of bleeding. Recent large RCTs and meta-analyses have demonstrated that a dual therapy based on a non-vitamin K oral anticoagulant and a P2Y_12_ inhibitor is superior to triple therapy based on VKA with DAPT to prevent bleeding [12,13,15,18,27,28]. However, a number of these trials also reported an increase, albeit not significant, of coronary ischemic events in case of aspirin discontinuation [13,15]. In the present meta-analysis, rates of MI or definite ST were also numerically higher in patients treated with background OAC and early discontinuation of aspirin, but did not reach statistical significance, despite substantially increased statistical power. In patients undergoing PCI, short DAPT regimens have been associated with a higher risk of stent thrombosis. However, this effect was mainly observed with first-generation DES and was no longer observed with second-generation DES, which was overwhelmingly used in the studies included in the present meta-analysis [6].

In our study, clopidogrel and ticagrelor were predominantly used as P2Y_12_ inhibitors based single antiplatelet therapy. Considering the significant proportion of patients presenting with inadequate response to clopidogrel therapy, as detected by platelet function or genetic testing, concerns were raised regarding its use as single antiplatelet therapy, particularly in patients without background OAC [29]. However, genotype or platelet function test-based strategies of P2Y_12_ inhibitors have not resulted in significant reduction of ischemic complications in dedicated trials, which further suggests that clopidogrel alone may be safely used in these patients [30]. Consistently, no significant interaction was found between the effect of early aspirin discontinuation and prolonged clopidogrel or ticagrelor single antiplatelet therapy in patients not requiring chronic OAC in our study. Recently, the Intracoronary Stenting and Antithrombotic Regimen: Rapid Early Action for Coronary Treatment (ISAR-REACT) 5 trial reported the superiority of prasugrel compared to ticagrelor in patients presenting with ACS for whom invasive evaluation is planned [31]. Prasugrel was only used in a minority of patients in the trials included in our meta-analysis and no conclusion can thus be drawn regarding its use as single antiplatelet therapy although, prasugrel has been used as single therapy in a few prior studies [32]. Of note, ticagrelor and not prasugrel is being evaluated as P2Y_12_ inhibitor based single antiplatelet therapy in other ongoing RCTs (ISRCTN84335288, NCT03447379, NCT03797651, and NCT02494895). Future trials are warranted to compare the performance of all commercially available P2Y_12_ inhibitors, used in clinical practice in the setting of early aspirin discontinuation.

Moreover, the optimal timing for aspirin discontinuation remains unclear. In all trials with patients presenting an underlying indication for chronic OAC, aspirin use was authorized during PCI and prior to randomization which usually occurred between four hours after arterial sheath removal up to 14 days after PCI/ACS [11,12,13,15,18]. Conversely, in trials enrolling patients without indication for chronic OAC, aspirin discontinuations occurred either at one month or at three months after randomization (mean weighted DAPT duration at 1.7 months). Of note, we did not find any significant interaction in the effect of aspirin discontinuation between one month and three months.

We report a significant interaction between background OAC and the risk of MI associated with early aspirin discontinuation. Although the role of pure chance cannot be excluded, as well as an issue with MI definitions as suggested by the sensitivity analysis, this effect might be real, reflecting the higher risk of OAC-treated patients, who are usually older and frailer than those without an indication for OAC.

### Limitations

Several limitations are to be acknowledged. Firstly, our findings are subject to the inherent limitations of the included RCTs, subsequent to the study design, follow-up, ischemic and bleeding events definitions, and events ascertainment. This is particularly the case for GLOBAL LEADERS where all events but new Q-wave MI were site-reported. Secondly, as we lacked patient-level data, we were unable to perform time-to-event analysis or to evaluate the safety and efficacy of the early aspirin discontinuation strategy according to clinical and procedural complexity. Thirdly, we included studies with heterogeneous inclusion/exclusion criteria, particularly regarding the underlying indication for chronic OAC, as well as differences in duration of antithrombotic treatment (i.e., overall duration for aspirin, P2Y_12_ inhibitors, or oral anticoagulation) which led to some degree of heterogeneity in the results. Fourthly, the risk of stent thrombosis decreases over time with new generation DES while we used a random effect modeling which assumes an equal chance of event at all time.

## 5. Conclusions

In patients on DAPT for an ACS or after a PCI, with or without an underlying indication for chronic OAC, a strategy of early aspirin discontinuation is associated with a significant reduction of major, non-major, and all bleeding, without detectable impact on mortality or ischemic risk.

## Figures and Tables

**Figure 1 jcm-09-00680-f001:**
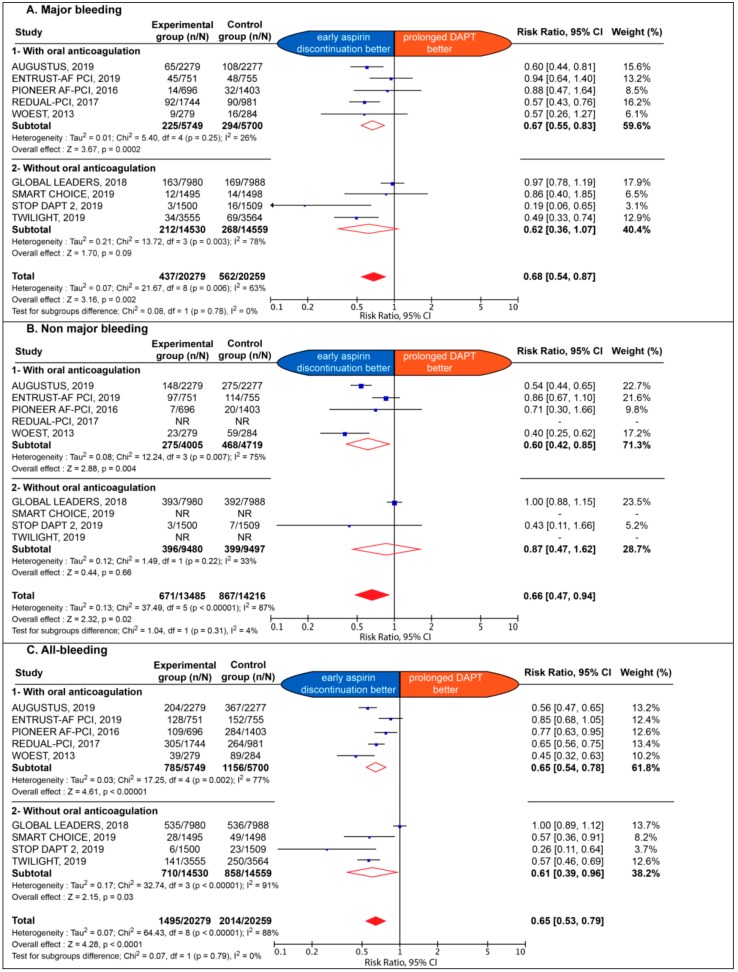
Estimate risk of major bleeding (**A**), non-major bleeding (**B**), and all bleeding (**C**). CI: confidence interval; DAPT: dual antiplatelet therapy; NR: not reported.

**Figure 2 jcm-09-00680-f002:**
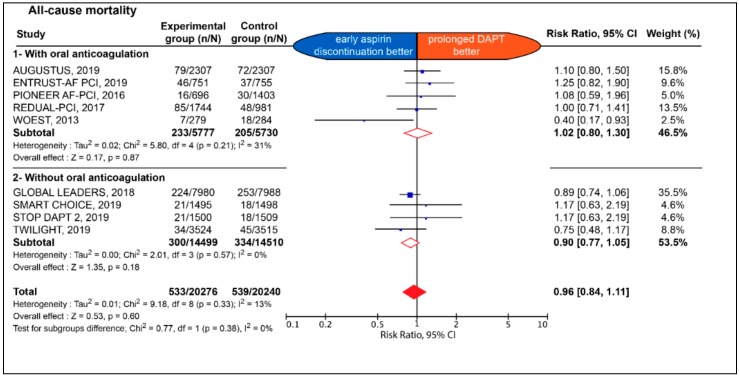
Estimate risk of all-cause death. CI: confidence interval; DAPT: dual antiplatelet therapy.

**Figure 3 jcm-09-00680-f003:**
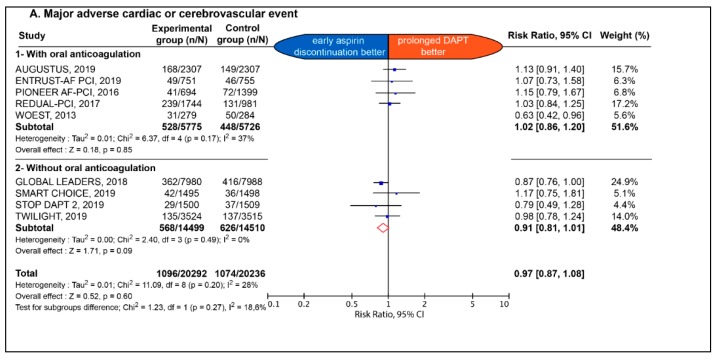
Estimate risk of major adverse cardiac and cerebrovascular events (**A**), myocardial infarction (**B**), definite stent thrombosis (**C**) and any stroke (**D**). CI: confidence interval; DAPT: dual antiplatelet therapy; NR: not reported.

**Table 1 jcm-09-00680-t001:** Characteristics of the included studies.

Study Publication Year Clinicaltrials.gov ID	Study Design	Main Inclusion Criteria	Main Exclusion Criteria	Sample Size	Follow Up	Early Aspirin Discontinuation			Standard of Care			Primary Outcomes
						Duration of Aspirin Therapy after Randomization	P2Y_12_ Inhibitors Use (Dosage)	OAC Agent	DAPT Duration	Antiplatelet Agents	OAC Agent	
WOEST [11] 2013 NCT00769938	Randomized, open label, multicentric, superiority, controlled trial	Indication for oral anticoagulation and PCI	>80 years, Prior ICH, cardiogenic shock recent major bleeding, thrombocytopenia	563	12 months	None after randomization	Clopidogrel 100% (75 mg)	VKA	1 to 12 months	Aspirin 80–100 mg; and Clopidogrel 75 mg	VKA	Safety: Any episode of bleeding (defined by TIMI, GUSTO or BARC classification
PIONEER AF-PCI [12] 2016 NCT01830543	Randomized, open label, multicentric, controlled trial	Non-valvular AF PCI with coronary stent implantation	Prior stroke/TIA, recent GI bleeding, severe CKD, anemia increase risk of bleeding contra-indication for OAC Active malignancy	2124	12 months	None after randomization	Clopidogrel 93.1% (75 mg), Ticagrelor 5.2% (90 mg bid), Prasugrel 1.7% (10 mg)	Rivaroxaban 15 mg or 10 mg	1, 6 or 12 months	Aspirin 75–100 mg, and Clopidogrel 75 mg, Ticagrelor 90 mg bid, Prasugrel 10 mg	VKA or Rivaroxaban 2.5 mg	Safety: Composite of: Major and minor TIMI bleeding and bleeding requiring medical attention
REDUAL-PCI [13] 2017 NCT02164864	Phase IIIb, randomized, open label, multicentric, non-inferiority, controlled trial	Non valvular AF Successful PCI < 120 h	Prosthetic heart valves, severe CKD, recent stroke, major surgery or GI bleeding	2725	14 months *	None after randomization	Clopidogrel 86.6% (75 mg), Ticagrelor 12.4% (90 mg bid)	Dabigatran 150 or 110 mg bid	1 month (BMS) 3 months (DES)	Aspirin < 100 mg and Clopidogrel 75 mg, Ticagrelor 90 mg bid	VKA	Safety: Time to event analysis of first major or clinically relevant non major ISTH bleeding
GLOBAL LEADERS [14] 2018 NCT01813435	Randomized, open label, multicentric, superiority, controlled trial	Clinical indication of PCI	Need for OAC, planned surgery, recent stroke, prior major bleeding	15,968	24 months	30 days	Ticagrelor 100% (90 mg bid)	N.A.	12 months	Aspirin 75-100 mg, and Clopidogrel 75 mg, Ticagrelor 90 mg bid	N.A.	Efficacy: Composite of all-cause death or non-fatal, new Q-wave myocardial infarction.
AUGUSTUS [15] 2019 NCT02415400	Multicentric, randomized with two-two factorial design, double blinded, non-inferiority, controlled trial	AF and recent PCI or ACS with planned used of at least 6 months of P2Y_12_	Other indication for OAC, severe CKD, prior ICH, coagulopathy, planned CABG	4614	6 months	None after randomization	Clopidogrel 93.2% (75 mg) Ticagrelor 5.9% (90 mg bid) Prasugrel 0.9% (10 mg)	Apixaban 5 mg or 2.5 mg bid or VKA	6 months	Aspirin 81 mg, and Clopidogrel 75 mg, Ticagrelor 90 mg bid, Prasugrel 10 mg	Apixaban 5 mg or 2.5 mg bid or VKA	Safety: major or clinically relevant non-major ISTH bleeding
STOPDAPT-2 [16] 2019 NCT02619760	Randomized, open label, multicentric, non inferiority, controlled trial	PCI with CoCr-EES without periprocedural complication	Need for OAC, prior ICH, use of other stents	3009	12 months	1 month	During 1st month Clopidogrel 60.2% (75 mg) Prasugrel 39.6% (10 mg) After 1st month Clopidogrel 100% (75 mg)	N.A.	12 months	Aspirin 81 to 200 mg and Clopidogrel 75 mg or Prasugrel 10 mg before 1 month, Followed by Clopidogrel 75 mg	N.A.	Safety and efficacy: Composite of cardiovascular death, MI, definite stent thrombosis, stroke and TIMI major and minor bleeding
SMART CHOICE [17] 2019 NCT02079194	Randomized, open label, multicentric, non inferiority, controlled trial	PCI with DES for ACS or stable CAD	Hemodynamic instability; active bleeding; recent DES implantation	2993	12 months	3 months	Clopidogrel 76.9% (75 mg) Ticagrelor 6.5% (90 mg bid) Prasugrel 0.7% (10 mg)	investigators choice	12 months	Clopidogrel 75 mg, Ticagrelor 90 mg bid, Prasugrel 10 mg	investigators choice	Efficacy: Composite of all-cause mortality, MI, stroke
ENTRUST-AF PCI [18] 2019 NCT02866175	Phase IIIb, Randomized, open label, multicentric, controlled trial	Non valvular AF and PCI procedure for stable CAD or ACS with success	Valvular or reversible AF, mechanical heart valve, severe CKD, major surgery planned, recent ischemic stroke, high bleeding risk	1506	12 months	None after randomization	Clopidogrel 92.7% (75 mg) Ticagrelor 6.5% (90 mg bid) Prasugrel 0.7% (5 or 10 mg)	Edoxaban 60 mg or 30 mg VKA	1 to 12 months	Aspirin 100 mg, and Clopidogrel 75 mg, Ticagrelor 90 mg bid, Prasugrel 5 or 10 mg	VKA	Safety: Composite of ISTH major and clinically relevant non-major bleeding Efficacy: Composite of CV death, stroke, systemic embolic event, MI and definite ST
TWILIGHT [19] 2019 NCT02270242	Phase IV, randomized, blinded-label, multicentric, superiority controlled trial	High risk patients with complex PCI †	Contraindication to aspirin or ticagrelor, STEMI as index event, need for chronic OAC, prior stroke, planned surgery or coronary revascularization	7119	12 months	None after randomization	Ticagrelor (100%)	N.A.	12 months	Aspirin 81–100 mg and Ticagrelor 90 mg bid	N.A.	Safety: Composite of BARC types 2, 3 or 5 bleeding

* mean follow-up † was defined as the association of at least one criterion among: age > 65 years, female sex, established CV disease, diabetes mellitus, chronic kidney disease, and at least one criterion among: multivessel disease, total stent length > 30 mm, thrombotic lesion, bifurcation, left main or proximal left anterior descending artery. PCI: percutaneous coronary intervention; DAPT: Dual antiplatelet therapy; OAC: oral anticoagulation; VKA: vitamin K antagonist; MI: myocardial infarction; AF: atrial fibrillation; TIA: transient ischemic attack; ICH: intracranial hemorrhage; GI: gastro-intestinal; CKD: Chronic kidney disease; ACS: acute coronary syndrome; CABG: coronary artery bypass graft; CAD: coronary artery disease; CV: cardiovascular; CoCr-EES: cobalt-chromium everolimus eluting stent; N.A.: not applicable; STEMI: ST segment elevation myocardial infarction.

**Table 2 jcm-09-00680-t002:** Baseline patients’ characteristics.

Study	Male Sex	Age (Years) *	Prior MI	Prior Coronary Revascularization	Diabetes Mellitus	Systemic Hypertension	Dyslipidemia	Active Smoking	ACS as iIdex Rvent	Type of Stent Used
WOEST	448 (79.6%)	EAD: 70.3 ± 7.0 DAPT: 69.5 ± 8.0	196 (34.8%)	PCI: 187 (33.2%) CABG: 130 (23.1%)	140 (24.9%)	386 (68.6%)	396 (70.3%)	102 (18.1%)	155 (27.5%)	None: 9 (1.6%) DES: 364 (64.6%) BMS: 175 (31.1%) Both: 14 (2.5%)
PIONEER AF-PCI	1581 (74.4%)	EAD: 70.4 ± 9.1 DAPT: 70.0 ± 9.1 and 69.9 ± 8.7	477 (22.5%)	-	624 (29.4%)	1571 (74.0%)	913 (43.0%)	141 (6.6%)	1096 (51.6%)	DES: 1403 (66.0%) BMS: 675 (31.8%) Both: 40 (1.9%)
REDUAL-PCI	2070 (76.0%)	EAD: 71.5 ± 8.9 and 68.6 ± 7.7 DAPT: 71.7 ± 8.9	699 (25.6%)	PCI: 912 (33.5%) CABG: 287 (10.5%)	993 (36.4%)	-	-	-	1375 (50.5%)	DES: 2251 (82.8%) BMS: 404 (14.9%) Both: 41 (1.5%) Other: 21 (0.8%)
GLOBAL LEADERS	12,254 (76.7%)	EAD: 64.5 ± 10.3 DAPT: 64.6 ± 10.3	3710 (23.2%)	PCI: 5,221 (32.7%) CABG: 943 (5.9%)	4038 (25.3%)	11,715 (73.4%)	10,768 (67.4%)	4169 (26.1%)	7487 (46.9%)	Biolimus A9-eluting stent: 94.6% of lesions; other stent in 6.5% of lesions
AUGUSTUS	3277 (71.0%)	EAD: 70.8 (64.4–77.3) DAPT: 70.6 (63.8–77.2)	-	-	1678 (36.4%)	4073 (88.3%)	-	-	2811 (60.2%)	-
STOPDAPT-2	2337 (77.7%)	EAD: 68.1 ± 10.9 DAPT: 69.1 ± 10.4	406 (13.5%)	PCI: 1032 (34.3%) CABG: 59 (2.0%)	1159 (38.5%)	2221 (73.8%)	2244 (74.6%)	710 (23.6%)	1148 (38.2%)	CoCr-EES
SMART CHOICE	2198 (73.4%)	EAD: 64.6 ± 10.7 DAPT:64.4 ± 10.7	127 (4.2%)	349 (11.7%)	1122 (37.5%)	1840 (61.5%)	1352 (45.2%)	791 (26.4%)	1741 (58.2%)	CoCr-EES: 1051 (35.1%) PtCr-EES: 967 (32.3%) BP-SES: 972 (32.5%)
ENTRUST-AF PCI	1120 (74.4%)	EAD: 69 (63–77) DAPT: 70 (64–77)	365 (24.2%)	PCI: 394 (26.2%) CABG: 95 (6.3%)	517 (34.3%)	1361 (90.4%)	981 (65.1%)	-	777 (51.6 %)	-
TWILIGHT	5421 (76.1%)	EAD: 65.2 ± 10.3 DAPT: 65.1 ± 10.4	2040 (28.7%)	PCI: 2998 (42.1%) CABG: 710 (10.0%)	2620 (36.8%)	5154 (72.4%)	4303 (60.4%)	1548 (21.8%)	4614 (64.8%)	Locally approved DES

* Age is provided as mean ± standard deviation or as median [IQR]; MI: myocardial infarction; ACS: acute coronary syndrome; DES: drug eluting stent; BMS: bare metal stent; STEMI: ST segment elevation myocardial infarction; NSTEMI: non-ST segment elevation myocardial infarction; CoCr-EES: Cobalt-chromium everolimus eluting stent; PtCr-EES: Platinum-chromium everolimus eluting stent; BP-SES: Sirolimus-eluting stent with biodegradable polymer; PCI: percutaneous coronary intervention; CABG: coronary artery bypass graft; EAD: early aspirin discontinuation group; DAPT: dual antiplatelet therapy group.

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
