# Peer review of "Early Aspirin Discontinuation Following Acute Coronary Syndrome or Percutaneous Coronary Intervention: A Systematic Review and Meta-Analysis of Randomized Controlled Trials"

_jcm, 2020, doi:10.3390/jcm9030680_

Round 1
Reviewer 1 Report
It is interesting manuscript, but I have some minor comments and suggestions.
Authors should modify the Abstract – The abstract should be a single paragraph and should follow the style of structured abstracts, but without headings.
Check your manuscript and provide space between words and brackets with references,
Second section should be called Materials and Methods,
In section Materials and Methods Authors should give all inclusion and exclusion criteria – what does it mean “main inclusion/exclusion criterion”?
In my opinion in section Materials and Methods should not give information for what was each Author responsible – this information should be presented in Authors Contribution,
There is no need to add dots after title of subsections,
Discussion is quite short,
References should be one more time check by Authors – follow the Instruction for Authors,
Author Response
It is interesting manuscript, but I have some minor comments and suggestions.
Authors should modify the Abstract – The abstract should be a single paragraph and should follow the style of structured abstracts, but without headings.
Response: We apologize for the error which was corrected. The abstract now appears as a single paragraph, without headings.
Check your manuscript and provide space between words and brackets with references,
Response: We have edited the manuscript so that there would a space between words and brackets with references.
Second section should be called Materials and Methods,
Response: We have edited the manuscript and the second section now appears as "Materials and Methods"
In section Materials and Methods Authors should give all inclusion and exclusion criteria – what does it mean “main inclusion/exclusion criterion”?
Response: We apologize for the unnecessary vagueness on our part. We have edited the section which now reads as the following "The inclusion criterion was a randomized controlled trial, with a clinical primary endpoint, comparing a strategy of early aspirin discontinuation (i.e. with aspirin placebo or no aspirin treatment) and P2Y12 inhibitors (clopidogrel, ticagrelor or prasugrel) continuation to a strategy of prolonged DAPT including aspirin and P2Y12 inhibitors, following ACS or PCI, in patients with or without indication for chronic OAC. No restrictions on follow-up or study size were applied. The exclusion criteria were observational study design (including single-arm pilot studies), non-English-language studies, editorial, letters, expert opinions, case reports or series and studies with duplicated data."
In my opinion in section Materials and Methods should not give information for what was each Author responsible – this information should be presented in Authors Contribution,
Response: We agree with the reviewer comment and have edited the manuscript consistently: “Two authors independently evaluated studies for eligibility and discrepancies were resolved by a third reviewer."
AND
"Relevant data elements, including baseline population and procedural characteristics were independently collected from each trial into a pre-specified structural dataset."
There is no need to add dots after title of subsections,
Response: Dots following title of subsections were removed
Discussion is quite short,
Response: We agree with the reviewer comment. However, we would like to respectfully emphasize that the discussion, in its present form and length, comprehensively covers the potential clinical impact of our results, while evaluating the consistency with prior published studies in the field and detailing the limitations of our work. We aimed at maintaining a concise style throughout the manuscript. We would also like to respectfully emphasize that the length of the present discussion (approximately a thousand words) is consistent (and even slightly longer) with prior published studies in the Journal of Clinial Medecine (doi: 10.3390/jcm8081143. or doi: 10.3390/jcm9020439.)
References should be one more time check by Authors – follow the Instruction for Authors,
Response: We apologize for the mistakes and have edited all references.
Reviewer 2 Report
To the Authors:
I enjoyed reviewing your manuscript, entitled “Early Aspirin Discontinuation Following Acute Coronary Syndrome or Percutaneous Coronary Intervention: A systematic Review and Meta-Analysis of Randomized Controlled Trials.” This study of 9 randomized controlled trials including 40,621 patients showed lower rates of major bleeding and non-major bleeding with early aspirin discontinuation compared to prolonged dual antiplatelet therapy with no difference in mortality, myocardial infarction, stent thrombosis, and stroke. The appropriate duration of dual antiplatelet therapy is a rapidly-evolving, highly-controversial topic in cardiology. The manuscript could be improved with attention to the following details:
Abstract. Background. This sentence is incomplete; it lacks a verb.
Introduction. The authors should be congratulated for including both US and European guidelines and trials.
Experimental Section. 2.3. Statistical analysis. Random effects modeling assumes that the chance of an event is equal at all times. However, we know that the likelihood of stent thrombosis decreases over time. This limitation should be noted in the Discussion.
Discussion. 4.1. Limitations. The included studies had different durations of aspirin, P2Y12, and OAC agents. This limitation must be acknowledged.
The manuscript will benefit from copyediting for standard English language usage and grammar.
Author Response
I enjoyed reviewing your manuscript, entitled “Early Aspirin Discontinuation Following Acute Coronary Syndrome or Percutaneous Coronary Intervention: A systematic Review and Meta-Analysis of Randomized Controlled Trials.” This study of 9 randomized controlled trials including 40,621 patients showed lower rates of major bleeding and non-major bleeding with early aspirin discontinuation compared to prolonged dual antiplatelet therapy with no difference in mortality, myocardial infarction, stent thrombosis, and stroke. The appropriate duration of dual antiplatelet therapy is a rapidly-evolving, highly-controversial topic in cardiology. The manuscript could be improved with attention to the following details:
Response: We thank the reviewer for this comment and the opportunity to improve our manuscript
Abstract. Background. This sentence is incomplete; it lacks a verb.
Response: We apologize for this mistake which was corrected. The background now reads as "The respective ischemic and bleeding risks of early aspirin discontinuation following an ACS or PCI remain uncertain."
Introduction. The authors should be congratulated for including both US and European guidelines and trials.
Experimental Section. 2.3. Statistical analysis. Random effects modeling assumes that the chance of an event is equal at all times. However, we know that the likelihood of stent thrombosis decreases over time. This limitation should be noted in the Discussion.
Response: We thank the reviewer for this revelant comment. We have added a limitation, as suggested by the reviewer: "Fourthly, the risk of stent thrombosis decreases over time with new generation DES while we used a random effect modeling which assumes an equal chance of event at all time."
Discussion. 4.1. Limitations. The included studies had different durations of aspirin, P2Y12, and OAC agents. This limitation must be acknowledged.
Response: We agree with the reviewer comment and have thus added the following limitation: "Thirdly, we included studies with heterogeneous inclusion/exclusion criteria, particularly regarding the underlying indication for chronic OAC, as well as differences in duration of antithrombotic treatment (i.e. overall duration for aspirin, P2Y12 inhibitors or oral anticoagulation) which led to some degree of heterogeneity in the results"
The manuscript will benefit from copyediting for standard English language usage and grammar.
Response: We apologize for the mistakes and have further proofread and edited the manuscript by a native English speaker.
Reviewer 3 Report
I had the pleasure to review the systematic review and meta-analysis: Early Aspirin Discontinuation Following Acute Coronary Syndrome or Percutaneous Coronary Intervention
The review addresses the early DAPT with Aspirin rather than P2 Y12 inhibitor discontinuation and the respective ischemic and bleeding risks associated with it. Authors have performed a review of randomized controlled trials (RCTs) comparing a P2Y12 inhibitor-based single antiplatelet strategy following early aspirin discontinuation to a strategy of sustained dual antiplatelet therapy (DAPT) in ACS or PCI patients requiring or not anticoagulation for other indications and have included 9 RCTs comprising 40,621 patients. Compared to prolonged DAPT, authors have found that major bleeding (2.2% vs. 2.8%; RR 0.68; 95%CI: 0.54 to 0.87; p=0.002; I²: 63%), non-major bleeding (5.0% vs. 6.1%; RR: 0.66; 95%CI: 0.47 to 0.94; p=0.02; I²:87%) and all bleeding (7.4% vs. 9.9%; RR: 0.65; 95%CI: 0.53 to 0.79; p<0.0001; I²: 88%) were significantly reduced with early aspirin discontinuation without significant difference for all-cause death (p=0.60), major adverse cardiac and cerebrovascular events (p=0.60), myocardial infarction (p=0.77), definite stent thrombosis (p=0.63), and any stroke (p=0.59). It is relevant for the clinicians especially in patients with underlying GI comorbidities that after an ACS or a PCI, early aspirin discontinuation could prevent bleeding events with no effect on the ischemic risk or mortality.
Author Response
The review addresses the early DAPT with Aspirin rather than P2 Y12 inhibitor discontinuation and the respective ischemic and bleeding risks associated with it. Authors have performed a review of randomized controlled trials (RCTs) comparing a P2Y12 inhibitor-based single antiplatelet strategy following early aspirin discontinuation to a strategy of sustained dual antiplatelet therapy (DAPT) in ACS or PCI patients requiring or not anticoagulation for other indications and have included 9 RCTs comprising 40,621 patients. Compared to prolonged DAPT, authors have found that major bleeding (2.2% vs. 2.8%; RR 0.68; 95%CI: 0.54 to 0.87; p=0.002; I²: 63%), non-major bleeding (5.0% vs. 6.1%; RR: 0.66; 95%CI: 0.47 to 0.94; p=0.02; I²:87%) and all bleeding (7.4% vs. 9.9%; RR: 0.65; 95%CI: 0.53 to 0.79; p<0.0001; I²: 88%) were significantly reduced with early aspirin discontinuation without significant difference for all-cause death (p=0.60), major adverse cardiac and cerebrovascular events (p=0.60), myocardial infarction (p=0.77), definite stent thrombosis (p=0.63), and any stroke (p=0.59). It is relevant for the clinicians especially in patients with underlying GI comorbidities that after an ACS or a PCI, early aspirin discontinuation could prevent bleeding events with no effect on the ischemic risk or mortality.
Response: We thank the reviewer for this comment
Reviewer 4 Report
I read with extreme interest the present article on the effect of early aspirin discontinuation in patients on DAPT after an ACS or a PCI. This review gives a significant contribution to the field, going beyond current guidelines. I would just suggest to add a reference for European guidelines, not only American GL.
Metanalysis included patients with ACS or a PCI, including CCS patients? If not it should be better addressed.
I would also ask to better explain the lack of anatomical, procedural, and clinical characteristics of patients, in order to evaluate the stent thrombosis risk factors such as stenting of left main, proximal LAD, suboptimal stent deployment etc.
Author Response
I read with extreme interest the present article on the effect of early aspirin discontinuation in patients on DAPT after an ACS or a PCI. This review gives a significant contribution to the field, going beyond current guidelines. I would just suggest to add a reference for European guidelines, not only American GL.
Metanalysis included patients with ACS or a PCI, including CCS patients? If not it should be better addressed.
I would also ask to better explain the lack of anatomical, procedural, and clinical characteristics of patients, in order to evaluate the stent thrombosis risk factors such as stenting of left main, proximal LAD, suboptimal stent deployment etc.
Response: We would like to thank the reviewer for this kind comment and relevant suggestions.
Following the reviewer comment we have also included the 2017 ESC focused update on dual antiplatelet therapy in coronary artery disease (doi:10.1093/eurheartj/ehx419.), as reference 3.
A total of 52.4% of the included patients presented with an acute coronary syndrome as an index event, meaning that 47.6% underwent percutaneous coronary syndrome in the setting of stable or chronic coronary artery disease.
Following the reviewer comment, we have included the online Table 4 (online material) which details the main procedural characteristics in each of the included study (i.e. arterial vascular access, number and localization of the treated lesion, total stent length). However, as we lack individual patient level data, we were unable to further evaluate the impact of an early aspirin withdrawal strategy according to the clinical and procedural complexity. We have acknowledged this limitation in the manuscript: "Secondly, as we lacked patient-level data, we were unable to perform time-to-event analysis or to evaluate the safety and efficacy of the early aspirin discontinuation strategy according to clinical and procedural complexity "
Round 2
Reviewer 2 Report
To the Authors:
Thank you for submitting a revision of your manuscript, entitled “Early Aspirin Discontinuation Following Acute Coronary Syndrome or Percutaneous Coronary Intervention: A systematic Review and Meta-Analysis of Randomized Controlled Trials.” The appropriate duration of dual antiplatelet therapy is a rapidly-evolving, highly-controversial topic in cardiology. The revised manuscript features improved English language usage and a more thorough discussion of the limitations of this meta-analysis. The authors should be congratulated for their fine work.